# Effect of Oil Dispersion on Lubricating Film Thickness Generation under Oil Droplet Supply Conditions

**Chenglong Liu** [1,2], **Wei Li** [1], **Feng Guo** [1,*], **Patrick Wong** [2] and **Xinming Li** [1]

[1] School of Mechanical & Automotive Engineering, Qingdao University of Technology, 777 Jialingjiang Road, Qingdao 266520, China; liuchenglong@qut.edu.cn (C.L.); shlw@qut.edu.cn (W.L.); lixinming@qut.edu.cn (X.L.)

[2] Department of Mechanical Engineering, City University of Hong Kong, 83 Tat Chee Avenue, Kowloon, Hong Kong 999077, China; meplwong@cityu.edu.hk

[*] Correspondence: mefguo@qut.edu.cn; Tel.: +86-13863945503

**Abstract:** Oil–air lubrication has proven to be very effective for high-speed bearings because the oil supply in the form of droplets can be precisely controlled. This work uses optical interferometry to study the mechanism of lubricating film formation in rolling point contact with oil droplet lubrication. The effect of a double oil drop pair, where two oil droplets are positioned in mirror images about the central axis of the lubricated track, is examined. The process by which pairs of oil droplets approach and lubricate a bearing contact is analysed. This study also covers the effect of multiple oil droplets supplied in a tailored or a random dispersion pattern. Additionally, the effects of oil viscosity, entrainment velocity, and droplet distribution on starvation are also investigated.

**Keywords:** point contact; oil droplets; film thickness generation; starvation; oil dispersion

## 1. Introduction

To reduce the usage of non-renewable petroleum-based oils, researchers have been trying to find better ways to lubricate machines by controlling or limiting the amount of oil applied [1]. It is crucial to optimize these measures as they may lead to oil starvation at the bearing contacts. In fact, only a very tiny layer of the lubricant is necessary to effectively separate the two contacting surfaces. Theoretically, the required amount of oil is quite small. Therefore, precise lubrication techniques such as oil–air lubrication [2–4], oil mist lubrication [5], and the use of super-lubricity additives [6] are employed, especially in high-speed machine components and harsh operating conditions. Depending on the method of delivering the lubricant, the supplied oil may manifest as discrete oil droplets. Moreover, the thin layer of oil on the lubrication track resulting from limited lubricant supply tends to generate discrete oil droplets [7], which directly influences the lubrication performance. Consequently, oil droplet lubrication has recently attracted extensive research attention. Additionally, in the case of LLS, the occurrence of starved lubrication is inevitable due to oil loss caused by the rolling motion of bearing elements [8]. Even when sufficient initial lubricant is provided, bearing failure can still occur, as exemplified by a notable case of wind turbine bearing failure, which was primarily attributed to severe oil starvation resulting from high-frequency reciprocating motion [9].

Oil starvation was initially examined by Wedeven et al. [10], who observed the starved film thickness by using optical interferometry, and they found that the inlet distance parameter can be used to describe the level of starvation. Pemberton and Cameron [11] found that in limited oil supply lubrication, side oil ridges are the primary sources for replenishing lubricating oil at the inlet of the elastohydrodynamic lubricated (EHL) contact area. Chevalier et al. [12] used the thickness of the oil layer at the bearing inlet area to define the degree of oil starvation and theoretically studied the influence of the oil supply condition on the lubricating oil film thickness. Chiu [13] studied lubricant replenishment behaviour

under starvation conditions and established a starvation lubrication model to explore the significance of lubricant replenishment. Guangteng et al. [14] and Kingsbury et al. [15] conducted optical EHL tests with different lubricants. They observed that under starved lubrication conditions, the lubricating film thickness decreases to a steady value in the range of tens of nanometres and is independent of speed. Cann et al. [16] experimentally studied the effects of oil volume, contact area size, viscosity, and speed on saturation. They defined a dimensionless parameter called the "starvation degree" to control the onset of starvation. Recently, van Emden et al. [17] studied the formation of cavitation bubbles at the exit of EHL contact during the startup process using a small amount of oil. Liang et al. [18] considered how centrifugal force and temperature influence starvation and explored how the oil pool shape affects oil–air lubrication. Obviously, having a large oil pool with a continuous supply of oil is important to produce an effective lubricating film thickness to separate the two surfaces. Nevertheless, even under oil starvation regimes, sufficient hydrodynamic pressure can be generated to keep the contact surfaces apart by an appropriate amount of oil replenished from the oil ridges on the two sides of the lubrication track.

In engineering practice, during high-speed rotation, a significant proportion of the lubricant is forced to both sides of the lubrication track and may not be able to flow back in time, such that there is not enough replenishment [19,20]. Sun et al. [21] studied the characteristics of the sound field distribution of full ceramic ball bearings under different oil supply conditions. They found that the bearing exhibited the minimum radiation noise, vibration, and temperature rise when operating under the optimal lubrication condition. Ge et al. [22] enhanced the lubrication performance of bearings by adding groove structures to the non-contact area. NTN [23] modified the position of the oil–air lubrication supply nozzle, changing it from the inner ring to the outer ring to avoid the influence of vortices. Maruyama et al. [24] explored the relationship between the supplied oil flow rate and oil film thickness under steady starved lubrication, aiming to increase the utilization rate of lubricating oil. Liang et al. [25] explored the replenishment of the oil layer out of the contact of the rolling bearing by using the laser-induced fluorescence technique and found that the starvation depended on the oil supply condition. By using CFD methods [26–29], researchers have studied the oil–air two-phase flow pass through the contact area and out of the contact area in a bearing. The oil transfer behaviours and flow pattern characteristics around the rolling element in the cavity inside the bearing were discussed in detail. The above studies on oil–air lubrication demonstrate that increasing the density of lubricant droplet distribution on the surface of tribo-pairs can effectively extend the service life and improve the operational performance of mechanical components.

The degree of starvation of oil droplet lubrication can be adjusted by changing certain parameters of the oil droplets, such as their size and viscosity. Zhang et al. [30] conducted numerical analyses to understand the lubrication behaviour of oil droplets in EHL contacts. They found that having a thick oil layer at the contact inlet is beneficial for lubrication. Li et al. [31] summarized the spreading behaviour of oil droplets of different viscosities and sizes around static point contacts and examined the oil film formation when oil droplets were entrained into the contact area. However, there have not been many studies on how well oil droplet lubrication works when multiple droplets are supplied.

The present work aims at investigating the lubrication performance when multiple oil droplets are supplied in different droplet dispersions. We examine the factors that control the spread of oil droplets and how they influence the formation of lubricating films at the contact area. Furthermore, this study explores the lubrication performance when oil droplets are supplied either in a tailored pattern or randomly.

## 2. Experimental Apparatus and Conditions

### 2.1. Apparatus and Specimens

A self-developed optical EHL test apparatus, as schematically shown in Figure 1, was used for observing the oil droplet lubrication process. The optical test rig consists of a steel ball loaded on a glass disc. The speeds of the glass disc and the steel ball can be set

independently, allowing for different SRR tests. In this work, two servo motors drive the disc and steel ball with the same speed by considering their different diameters. The film thickness measurement utilizes the principle of optical interference. Red (wavelength 640 nm) laser and green (wavelength 525 nm) laser were employed. The highly polished steel ball has a diameter of 25.4 mm and a roughness Ra of 10 nm. The working surface of the disc was coated with a thin chromium (Cr) layer (10 nm) covered by a silica ($SiO_2$) layer (150 nm) [32], having a roughness Ra of 8 nm, to facilitate the interferometry measurements of the lubricating film. The interferometry images were obtained by multiple reflections and refractions of the red and green lasers, which were then captured by a high-speed camera (maximum capture frame rate: 1000 fps) after magnification using a microscope.

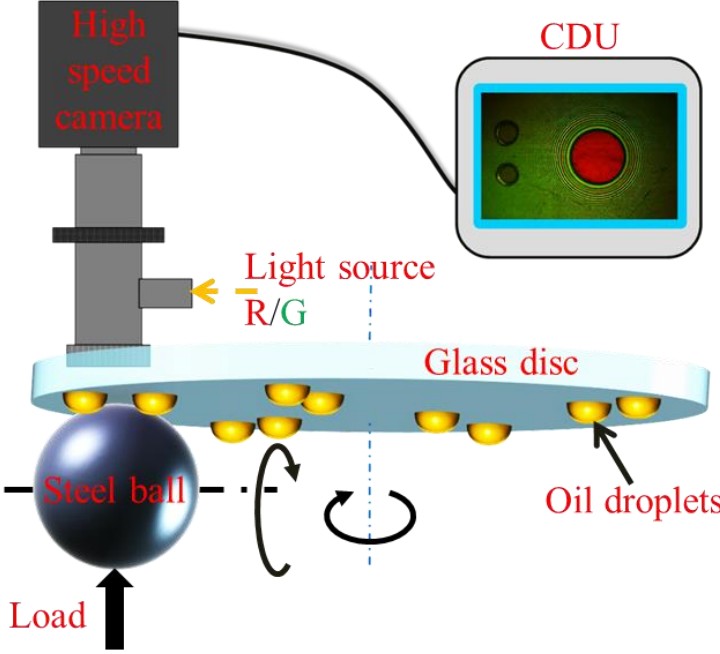

**Figure 1.** Schematic diagram of the test apparatus.

In practice, oil droplets are supplied and deposited on the surface in the form of sporadic and discrete droplets, as shown in the high-speed photographs in Figure 2. When multiple oil droplets approach a bearing contact, the interaction of the oil droplets may affect subsequent lubricating film generation. As shown in Figure 2, the oil droplet in the dotted blue cycle merged into one large droplet during the entire process of passing through the contact area and contributed to the film formation. However, the oil droplet in the dotted red cycle did not participate in the lubrication of the contact area. Therefore, the effect of oil droplet dispersion on lubrication deserves study in detail.

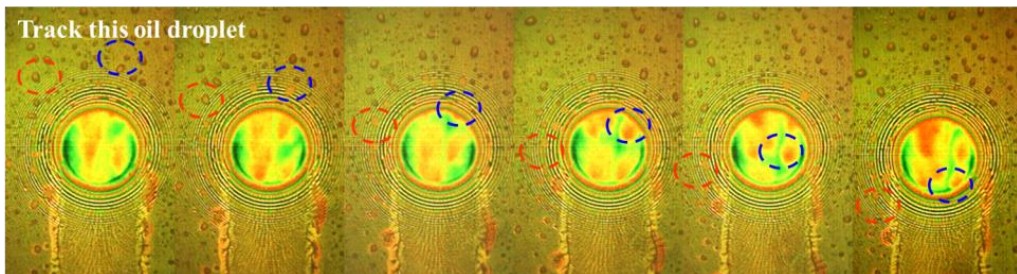

**Figure 2.** Oil droplets settlement on surface of oil–air lubrication, 32# engine oil, 30 N, 10 mm/s, 0.25 mL/h, 2 bar, droplet in the blue dotted circle contributed to the film formation, droplet in the red dotted circle went around the contact area.

The lubricants used in the experiment and their related properties are listed in Table 1. All the tests were run under pure rolling conditions, which were activated by a servo motor directly connected to the central shaft of the glass disc. A high-speed video camera was used to record the moving process of the oil droplets entering the contact region.

**Table 1.** Properties of the lubricants.

| Lubricant | Viscosity $\eta$ (Pa·s @ 20 °C) | Density $\rho$ (kg·m$^{-3}$ @ 20 °C) | Refractive Index | Contact Angle $\theta$ (°) |
|---|---|---|---|---|
| PAO 20 | 0.241 | 0.845 | 1.469 | 19.5 |
| PAO 40 | 0.810 | 0.827 | 1.465 | 29.5 |
| PAO 100 | 1.800 | 0.847 | 1.405 | 35.0 |
| 5P4E | 5.410 | 1.205 | 1.630 | 40.0 |

The volume of an individual oil droplet can be inferred by using its base diameter and contact angle, which are calculated using Equation (1), assuming that the droplet is in the form of a perfect circular segment. In the experiment, oil droplets were applied to the disc surface using the tip of a sharp pin to precisely control the droplet size and generate oil droplets with the desirable dimensions. The dichromatic interference intensity modulation approach was used to reconstruct the film profiles based on these interference images. Experiments were performed at ambient temperature ($20 \pm 1$ °C). Before each test, the ball and the disc were both cleaned with ethanol and acetone for 10 min and then dried with dry nitrogen.

$$V = \frac{\pi d_{\text{ini}}^3}{24} \frac{(1 - \cos\theta)^2 (2 + \cos\theta)}{\sin^3\theta} \tag{1}$$

### 2.2. Arrangements of Oil Droplets

In this work, the oil dispersion is defined by the distance $L$ between the inner edges of an oil droplet pair, as illustrated in Figure 3. When the oil droplets have not yet entered the contact area, they move with the glass disc at a speed of $u_d$. Once the oil droplets contact the surface of the rolling element, they begin to spread. The spreading process is governed by the squeeze action and the capillary effect of the tiny gap between the two rolling surfaces. At the moment when the oil droplets are entrained into the contact area, they move in the $x$ direction at a speed of $u_e$. The squeeze action results in the increase in the spatial area of the droplets and capillary forces drive the droplets towards the centre of the lubrication track. In the whole process, droplets lubricate the surface as it contacted the steel ball. Thus, this moment is set as the initial time ($T = 0$ s).

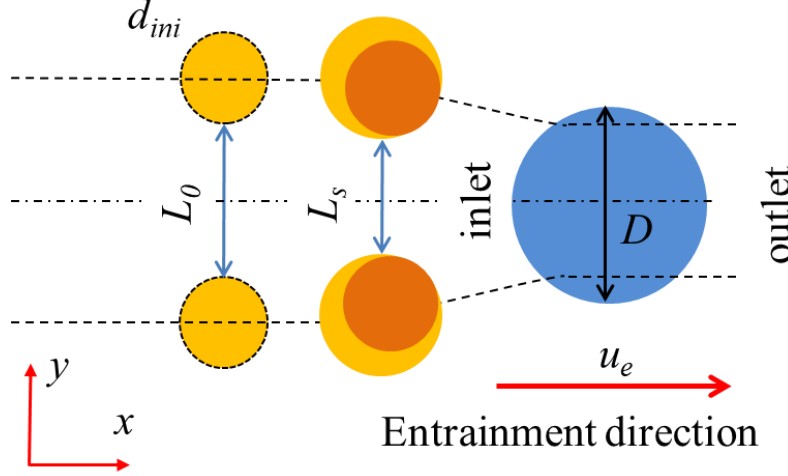

**Figure 3.** Schematic illustration of droplets transportation process.

The lateral distance of the droplets is measured from their edges. The spreading process is defined as the change in the lateral distance between the moments when the droplets come in contact with the ball surface and when they enter the contact region. The change in distance between the two moments ($L_o - L_s$) represents the total lateral movement of the droplets before they enter the bearing contact region. $L_0$ represents the initial distance between the inner edges of two oil droplets, while $L_s$ denotes the distance between their inner edges at the moment when their edges reach the contact area. After the oil droplets completely enter the contact area, the EHL film is formed. Under pure rolling conditions, the change in the shape of the oil pool and the film formation in the contact area during the entire lubrication process are recorded by a high-speed camera.

## 3. Results and Discussion

### 3.1. Lubrication under Single and Double Oil Droplet Supply

The characteristics of EHL film formation with one and two oil droplets were evaluated. The tests were conducted under the conditions of a constant rolling speed of 4 mm/s and a load of 30 N. PAO100 was used as the lubricant. The process of oil droplet(s) passing through and lubricating the contact area is shown in Figure 4.

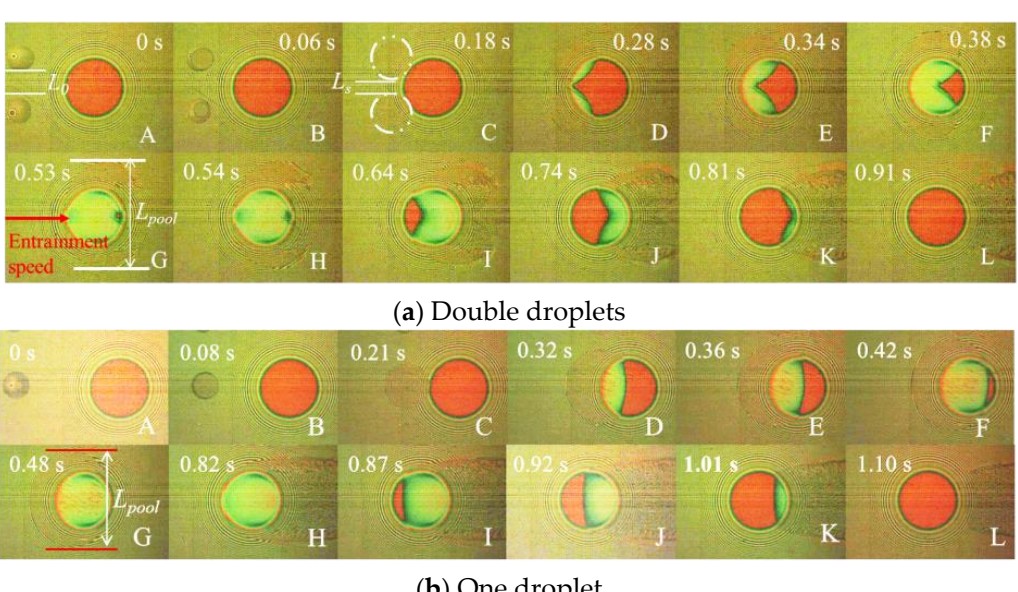

(**a**) Double droplets

(**b**) One droplet

**Figure 4.** EHL film formation with oil droplets under pure rolling, the initial diameter of oil droplet, $d_{ini}$ = 160 μm, $u_e$ = 4 mm/s, $w$ = 30 N, PAO100, (**a**) double oil droplets (the initial distance between two oil droplets, $L_0$ = 180 μm); (**b**) one droplet.

Two tiny oil droplets were placed on each side of the running track, as shown in photo A of Figure 4a. The droplet pair approached the contact area from the left in the photos. The lateral distance between the two droplets, $L_0$, started to decrease when the droplets came into contact with the ball surface. Photo B shows the shift of the two circular patches towards the centre of the Hertz contact, where the oil droplets meet the ball surface. The shift of the circular patches illustrates the spreading direction of the oil droplets. The build-up of lubricating films along the centreline of the contact area lagged behind those on the two sides. Photos D to H show that lubricating films initially developed from the two sides of the contact. By the time reaching 0.53 s (photo G), the lubricated contact acquired largely classical horseshoe-shaped films and this instant is referred to as the optimal lubricated instant. Photo G shows that there exist small crescent-shaped films at the inlet and outlet as the footprints of the merging of the lubricating films from the two sides, as shown in D to F. Obviously, if continuous oil droplets are supplied and merged together in the contact region, the complete full film shown in photo G can be sustained. The width of the oil pool ($L_{pool}$) did not change significantly during the departing process

(from G to L) since no oil was replenished in the test. The total duration for maintaining a complete full film that covers the contact area, referred to as the effective lubricated time *T*, depended on how effectively the two oil droplets merged together.

Figure 4b depicts the lubricating performance of a single oil droplet with a diameter similar to that of either droplet in Figure 4a. The single oil droplet can provide a longer effective lubricated time (0.34 s from G to H in Figure 4b compared to 0.01 s from G to H in Figure 4a). Building complete full films in the contact area with a single oil droplet transported along the central line was easier than in a contact with two droplets. The reason is that a single droplet spread evenly in contrast to two oil droplets, which require more time to merge together. Furthermore, at the optimal lubricated instant (photo G in Figure 4), the lubricated contact area creates the classical horseshoe-shaped EHL film if it is supplied with enough lubricant. Figure 5 shows the film profile of one and two oil droplets supplied at the optimal lubricated instant. The central film thickness of one oil droplet was higher than that of two oil droplets, but using two oil droplets distributed more evenly results in a smaller difference between the central and minimum film thicknesses.

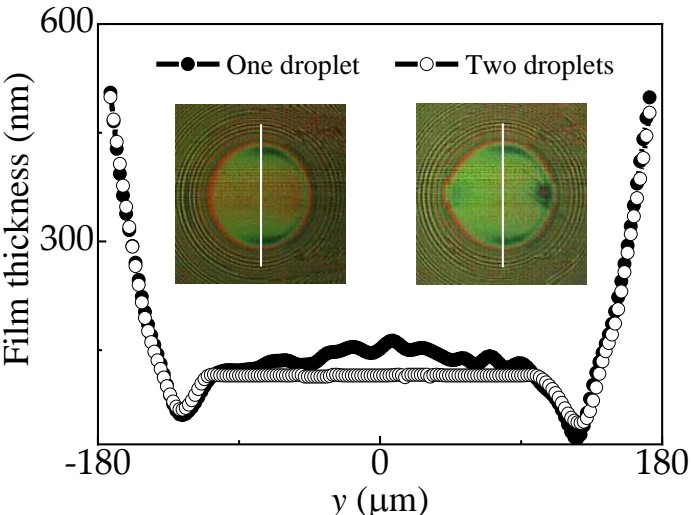

**Figure 5.** Lateral film profiles at *x* = 0 (white line marked in the images) at the optimal lubricated instants of Figure 4.

Comparing the results in Figures 4 and 5, it is evident that lubrication with two oil droplets is more prone to starvation. This clearly depends on the degree of the spreading and merging of the two oil droplets before entering the contact. However, laying oil droplets precisely along the central axis of the rolling track, as shown in Figure 4b, is difficult, even with the finest oil–air lubrication system. Oil droplets are more likely to be located away from the centreline. The ability of two neighbouring oil droplets to merge when they approach the contact region affects the degree of starvation, which can also be observed in Figure 2.

### 3.2. Effect of Oil Viscosity

Viscosity plays a crucial role in assisting the formation of the lubricating film. However, high viscosity diminishes the merging ability of oil droplets. Figure 6a illustrates oil droplets of different viscosities (PAO40 and PAO100) passing through the Hertz contact area. All the tests used oil droplets with an initial diameter of about 150 μm and an initial gap size of about 170 μm. Figure 6b shows the lateral distance between the two oil droplets at the inlet of the contact area $L_s$, the central film thickness at the optimal lubricated instant $h_c$, and the effective lubricated time *T*. $h_c$ increased while *T* decreased with the increase in viscosity, which is also shown in the complete full film lubrication process (from C to D) in Figure 6a. The spatial area they covered increased as the viscosity decreased. The two oil droplets of low viscosity moved closer at the instant shown in photo A of Figure 6a.

The lower viscosity PAO 20 oil droplets provided a better lubricating effect, as illustrated by its long effective lubricated time in Figure 6b.

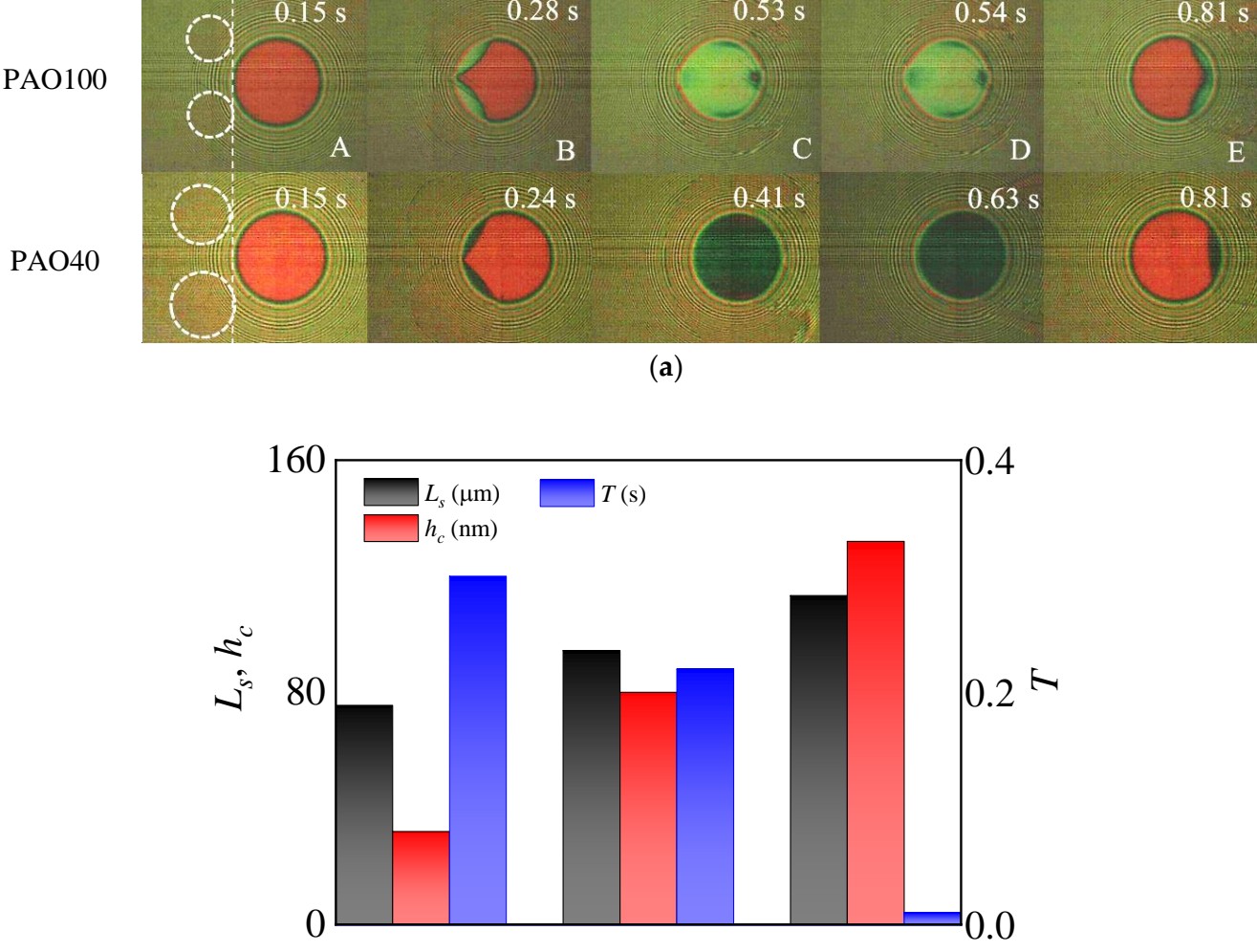

**Figure 6.** Differences in lubrication effect of oil droplets with different viscosities, $u_e$ = 4 mm/s, $d_{ini}$ = 150 μm, $L_0$ = 170 μm, 30 N, (**a**) Lubrication process over time; (**b**) Comparison of $L_s$, $h_c$, and $T$ for three oils, $L_s$ denotes the distance between their inner edges at the moment when their edges reach the contact area, $h_c$ is the central film thickness and T represents the duration of contact area with a complete film.

Figure 7 shows the temporal change in the lateral distance between the oil droplets ($L_s$) for three different oil samples. The spreading process started once the oil droplet touched the ball surface. The initial gap between the two oil droplets was 170 μm, and it gradually decreased over time. The total duration shown in Figure 7 is 0.16 s, representing the time taken for the oil droplets to go through the entire spreading process. As the oil droplets reached the inlet edge, their spatial area had significantly increased and they had come closer to the centreline, as shown in the insets of Figure 7 which show the starting and end moments of the spreading process. The oil droplet pairs of low viscosity came closer together at a higher rate compared to those of high viscosity. PAO 20 oil droplets moved fastest towards the centreline among the three because of their low viscosity.

Therefore, less viscous oils are preferred because they have better mobility for merging and converging actions.

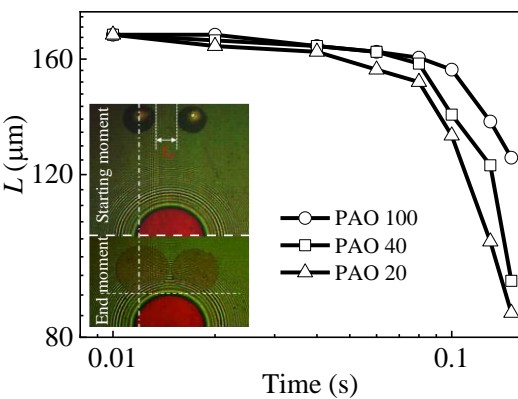

**Figure 7.** Lateral distance vs. time of double oil droplets with different viscosities, $d_{ini}$ = 150 μm, $L_0$ = 170 μm, 30 N.

### 3.3. Effect of Speed

Figure 8 presents the film building process at two different speeds. When the oil droplets passed through the contact area at a high speed (16 mm/s), the oil volume was not enough to create a complete full film for the entire bearing contact even at the instant of optimal lubrication (0.12 s, Figure 8b). Furthermore, the shape of the film boundary on the centreline at the contact region was more concave than that of the lower speed. This was because there was not enough time for the two oil droplets to merge.

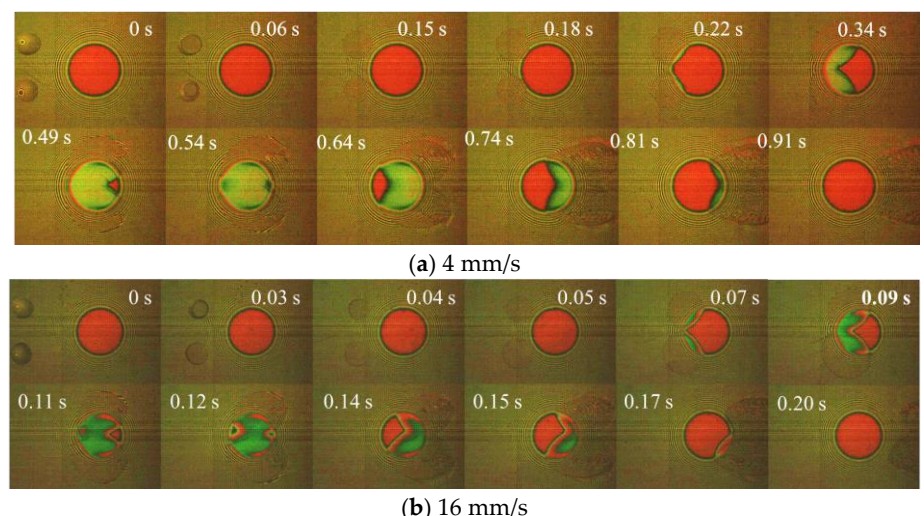

**Figure 8.** Effect of velocity with double oil droplets supply, $d_{ini}$ = 150 μm, $L_0$ = 170 μm, PAO100, 30 N, (**a**) 4 mm/s; (**b**) 16 mm/s.

In contrast, the merging of the two oil droplets at low speeds was good enough to achieve full lubricating films for the entire contact region. The gap ($L_s$) between the two oil droplets was smaller when they reached the inlet because they had adequate time to merge at low speeds. At the optimal lubricated instant (0.54 s, Figure 8a), the entire contact area was covered with oil. At low speeds, the oil droplets had more time to spread and were noticeably affected by capillary forces. The film boundary was less concave at the two ends of the centreline of the contact area. It can be inferred that the two oil droplets could have formed a single oil pool if the speed had been low enough.

Figure 9 displays the film thickness at different velocities under double oil droplets lubrication. The effective lubricated time $T$, which is the duration for maintaining a

complete full EHL film, decreases as the entrainment speed increases, as depicted in Figure 9. The central film thickness at the optimal lubricated time gradually increases with velocity. Figure 10a displays interferometry images of optimal film shapes at different speeds. The degree of starvation increases with speed. Therefore, although the central film thickness of 32 mm/s is higher than that of 4 mm/s, the EHL films of 32 mm/s remain incomplete and cannot cover the entire contact region. Figure 10b shows the optimal lubricating film thickness with a large lateral distance of two PAO oil droplets at different speeds. Overall, the film thickness increases with speed. However, the oil droplets cannot merge enough at 32 mm/s. The merging area has the minimum film thickness at the centreline of the EHL contact area. As a result, the resultant lubricating film, which is largely affected by the supply of oil droplets, is significantly lower than the corresponding film thickness under fully flooded conditions, as shown in Figure 9. At high speeds, building a complete full lubricating film that covers the entire contact region needs a high replenishment rate of oil droplets.

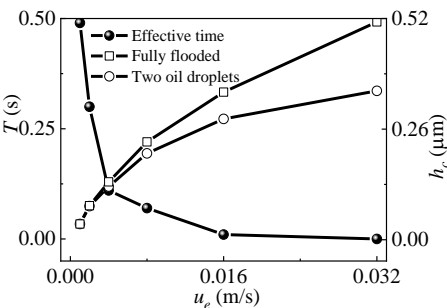

**Figure 9.** Effective lubricated time $T$ and central film thickness $h_c$ at the optimal lubricated instant at different velocities, $d_{ini}$ = 150 μm, $L_0$ = 170 μm, PAO 100, 30 N.

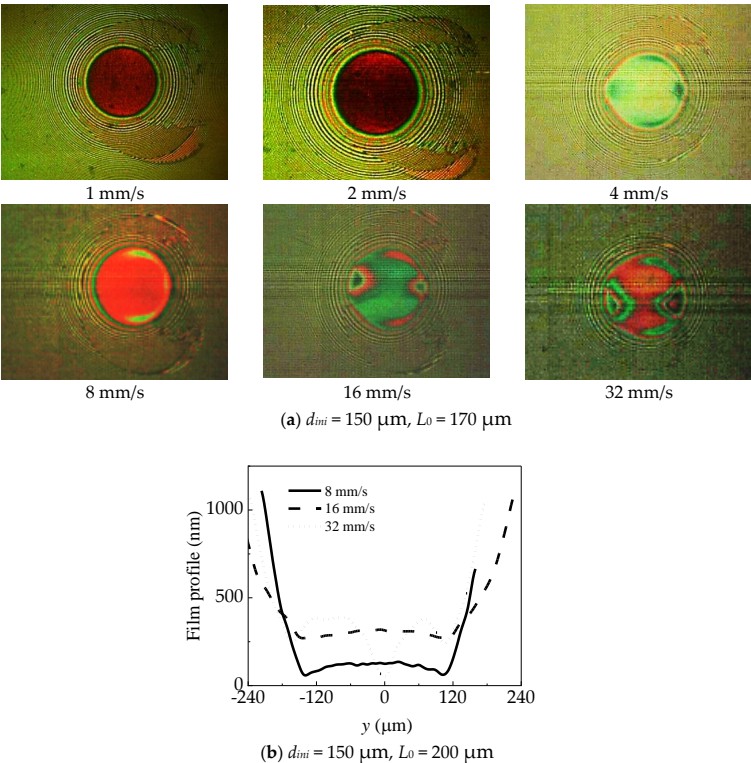

**(a)** $d_{ini}$ = 150 μm, $L_0$ = 170 μm

**(b)** $d_{ini}$ = 150 μm, $L_0$ = 200 μm

**Figure 10.** Interference images and film profile (at $x$ = 0) at optimal lubricated instance of different speeds, **(a)** $d_{ini}$ = 150 μm, $L_0$ = 170 μm; **(b)** $d_{ini}$ = 150 μm, $L_0$ = 200 μm.

### 3.4. Effect of the Initial Lateral Distance

Experiments were conducted with PAO 100 for three different initial lateral distances ($L_0$). All oil droplets had the same volume. The film shapes were captured at the optimal lubricated instant, and the central film profiles ($x = 0$) were plotted in Figure 11a. The central film thickness decreases as the initial distance increases. In the case with the largest initial gap ($L_0 = 289$ μm), complete full film cannot be achieved, as shown in Figure 11a. Increasing the distance leads to a reduction in the central film thickness, but the minimum film thickness does not change significantly, as shown in Figure 11b.

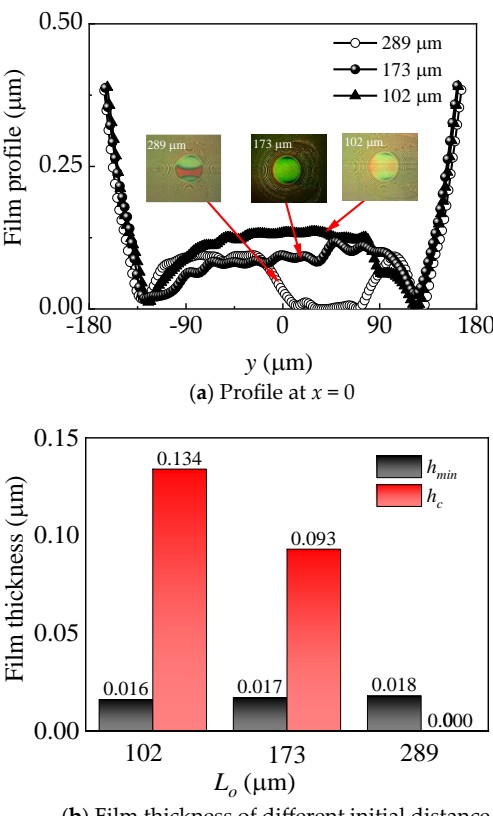

(**a**) Profile at $x = 0$

(**b**) Film thickness of different initial distance

**Figure 11.** Film thickness obtained at the optimal lubrication instant, $d_{ini} = 195$ μm, $u_e = 4$ mm/s, PAO100, 30 N, (**a**) Profile at $x = 0$; (**b**) Film thickness of different initial distance.

### 3.5. Effect of Oil Droplet Distribution

Oil–air lubrication relies on a series of oil droplets to attain continuous and favourable lubrication. The distribution of the oil droplets is crucial for achieving full film lubrication. Before the test, a uniform layer was used with a steel ball loaded with 30 N (Hertz diameter is 333 μm) to distribute 0.5 μL 5P4E oil along three lubrication tracks to cover the whole lubrication track, with a mean radius of 62 mm. After 10 min, the average layer would separate to small droplets by the dewetting effect. The orderly oil supply used a definite volume (42 pL) of oil droplets with a diameter of 82.5 μm and a spacing size of 167 μm in the $x$ direction and 195 um in the $y$ direction to arrange on the lubrication track, with a mean radius of 62 mm. The total volume on the lubrication track was also 0.5 μL.

Figure 12a shows the film thickness interferometry images of the same volume 5P4E lubricating oil with casual and orderly oil supply (with the same volume of oil supplied on a track with a width of 1 mm). The degree of dispersion of oil droplets in the tailored distribution is smaller compared to the casually distributed oil droplets. When casual oil droplets pass through the contact area, those distributed far from the contact area do not effectively entrain into the contact area and do not contribute to lubrication. Furthermore, individual droplets in casual distribution are smaller when compared to the tailored distribution. This size difference results in a shorter time for spreading and merging

between different oil droplets under casual droplet supply. Consequently, the oil film profile becomes uneven, as shown in Figure 12b. In contrast with the tailored distribution of oil droplets, merging occurs more efficiently, resulting in the formation of a uniform oil film with the typical EHL horseshoe shape. Figure 12c shows the different curves of central film thickness versus speed under casual and tailored droplet supply conditions. For higher merging efficiency, orderly distributed droplets result in a higher film thickness than the casual distribution. This indicates that a higher distribution concentration of lubricant droplets leads to better film formation performance. The tailored droplets improved the concentration of distribution, which can replenish more oil to merge together at the inlet area. However, under the casual oil droplets supply, the film thickness at the contact area is inadequate to form a uniform film profile for more droplets bypassing the contact area.

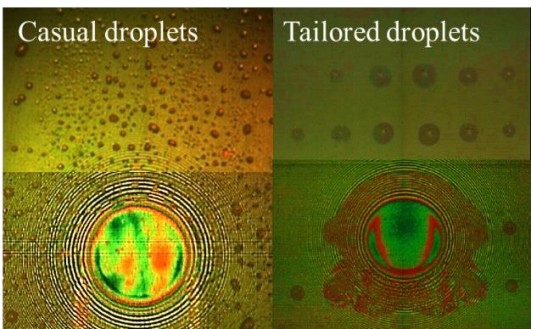

(**a**) Interferometry images under droplet supply, 10 mm/s

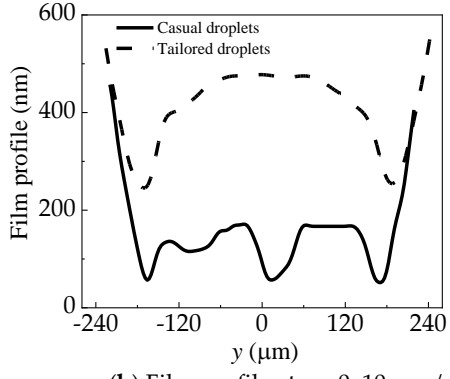

(**b**) Film profile at $x = 0$, 10 mm/s

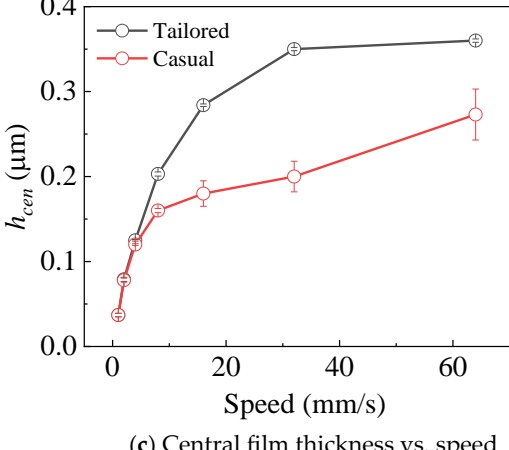

(**c**) Central film thickness vs. speed

**Figure 12.** Film profile at the optimal lubrication instant, total volume = 0.5 μL, 5P4E, 30 N, (a)Interferometry images under droplet supply, 10 mm/s; (**b**) Film profile, 10 mm/s; (**c**) Central film thickness vs. speed.

Based on the previous discussions, optimizing the parameters for oil droplet distribution brings benefits to film formation and increases the utilization rate of the lubricant. Figure 13 illustrates the differences in film thickness generated between the optimized oil droplet supply condition and a limited quantity oil supply condition. In the tailored distribution supply, four PAO100 oil droplets with a diameter of 150 μm were deposited along the lubrication track, with a spacing size of 170 μm in the *x* direction and 340 μm in the *y* direction. The film thickness curve, indicated by the red line, is plotted against velocities ranging from 1 to 32 mm/s. Referring to Table 1 for the contact angle of the lubricating oil, each oil droplet had a volume of 215 pL. The diameter of the lubricating track was 62 mm, and the amount of lubricant required for a complete track arrangement was 0.5 μL. According to the total amount of oil droplets, 0.5 μL and 2 μL PAO100 lubricating oil were uniformly arranged on the track at one time for comparison.

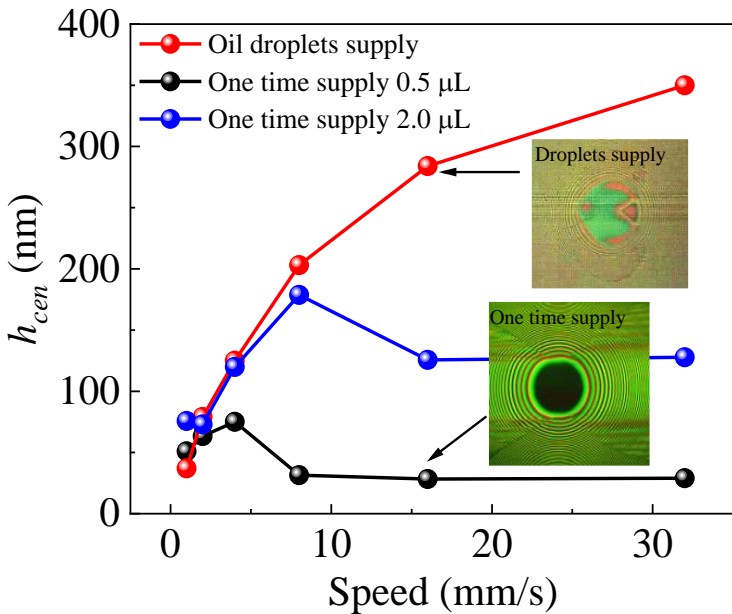

**Figure 13.** Central film thickness vs. speed at different oil supply conditions, PAO100, 30 N.

Under these same volume supply conditions, the film thickness curve demonstrates the advantages of the oil droplet supply mode. By selecting appropriate oil droplet distribution parameters, the amount of lubricant entering the contact zone can be increased compared to the uniform oil supply condition, where a significant portion of the lubricating oil flows around and leads to wasted oil. Simultaneously, improving the utilization rate of the lubricant also alleviates the viscous resistance caused by excessive oil supply. As depicted in the figure, even when the single oil supply reaches four times (2 μL), the lubricating oil film still fails to reach the film thickness observed under the oil droplet supply conditions.

## 4. Conclusions

Oil spreading under tailored oil droplet supply was studied. The spreading parameter, which reflects the effectiveness of spreading and oil droplet merging, was analysed through experiments. The following conclusions were drawn from the experimental results:

(1) The arrangement of oil droplets on the tribo surface affects the film building process: a single oil droplet results in a higher central film thickness, but when double oil droplets are distributed properly, it leads to a smaller difference between the central and minimum film thicknesses.

(2) Under multiple oil droplet supplies, creating a convergent oil pool is easier with low-viscosity oil than with high-viscosity oil. This leads to the formation of a full film in the contact area. In contrast, viscous oil droplets result in poor lubricant supply.

(3) The merging of two oil droplets is limited with high entrainment speeds. Capillary force and spreading effects are reduced at high speeds, weakening both the effective lubrication time and film formation capacity due to starvation. However, at high speed, more oil volume of the merging oil pool was used for film formation with a higher efficiency, other than removal to the non-contact area at lower speed. The relationship between speed and volume removal needs to be investigated in detail.

(4) Tailoring the distribution of oil droplets has greater advantages in forming the desired film thickness when compared with a random supply of oil droplets.

(5) Regarding the capacity of lubricating film formation, the optimal oil droplet supply condition is superior to the LLS condition with a uniform oil layer.

**Author Contributions:** Conceptualization, F.G. and P.W.; Investigation, C.L. and W.L.; Methodology, C.L., W.L., F.G. and X.L.; Validation, F.G., X.L. and P.W.; Writing—original draft, C.L.; Writing—review and editing, F.G. and P.W. All authors have read and agreed to the published version of the manuscript.

**Funding:** This research is funded by the Research Grants Council of Hong Kong (No. CityU 11215021), the National Natural Science Foundation of China (No. 52175173), the National Natural Science Foundation of China (No. 52205201), and the Youth Foundation of Shandong Natural Science Foundation (ZR2022QE027).

**Data Availability Statement:** The data used to support the findings of this study are included in the manuscript.

**Conflicts of Interest:** The authors declare no conflict of interest.

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
