# Peer review of "Effect of Oil Dispersion on Lubricating Film Thickness Generation under Oil Droplet Supply Conditions"

_lubricants, doi:10.3390/lubricants11120512_

Round 1

Reviewer 1 Report

Comments and Suggestions for Authors

The authors present an extensive study on the lubrication of EHL contacts with single droplets. The authors are to be congratulated for the great design and systematic structure.

The following minor points would make the paper even better and increase benefit for the reader.

P2 Line 57 “flow back if there IS not enough”

P3 Line 95 Cr layer and SiO2 layer. Formulation unclear. Both layers are present? What does thin mean (i.e., how thin?)

How do the authors deal with a thickness difference of the SiO2 layer, i.e., a layer variation around the circumference? PCS devices utilise a measurement at only one point, i.e., the point of measuring the “zero layer thickness”. Do the authors have such a system in place? Or are the authors sure that the SiO2 layer variation is small?

P3 Line 104 “blue dashed box” isn’t it a circle?

P3 Line 107 The sentence: “Therefore, the effect of oil droplets dispersion on lubrication are deserved to investigate deeply.” is unclear. What do the authors want to say? Please rewrite and clarify.

P3 Line 114: “pure rolling conditions … servo motor”. If the axis of the drive is parallel to the surface of the disc (as in Fig 1), is there not a tiny amount of spin (i.e., slip) induced? Was this considered in some way? How big is this?

P3 Line 116: “The oil droplet desirable volume could be detected by their base diameter and contact angle, calculated by Eq. (1).” Can the authors elaborate on this? Does this mean the authors controlled the volume (sounds like it from the word desirable) or is the volume of one droplet a fluid property (function of viscosity, contact angle, etc) ? If the volume was chosen then how was the “desirable” volume determined? If not controlled but actually different for the different lubricants is this mentioned somewhere? Why not always supply the same volume?

P4 Eq 1 and Table 1 maybe mention the theta is the contact angle (in table at least)

P8 Figure 9:

  • Can the authors elaborate how the time 0s was set in all their experiments? This is also important for Fig4,6,8.

  • The authors give the effective Lubricated times T in s. And define it as the time for maintaining a full film covering the contact area. Can the authors elaborate? If the contact is covered at a lower than full film thickness but no (red) areas are observed then it counts as effective T. Is this correct?

  • This time decreases with increasing velocity. The reached film thickness also increases. Can the authors discuss a bit more detailed their thoughts on the two effects: Lower film thickness (lower volume forming the full coverage) and higher velocity thus higher speed that this volume is removed from the contact. Might there be a relation with speed so that the observed T can be non-dimensionalised? So that the authors might be able to show the effect of lower build up due to the droplet spread and the higher removal of oil from contact uncoupled?

The conclusions give a good summary of the research findings. It might be beneficial to also give a summary regarding effective Time i.e, make point 3 a bit bigger. That is give application engineers an understanding what that means for design of lubrication systems.

Great read. The reviewer thanks the authors for the enjoyable paper! 

Reviewer 2 Report

Comments and Suggestions for Authors

The authors presented an in-depth study on oil droplet EHL. It is a detailed and interesting work that can be accepted for publication.

The authors can consider the following minor points:
1.
Thickness of the layers must be mentioned
2.
Flowback is to be discussed if the authors have experienced it.
3.
What about the contribution due to the motors? Please mention them.
4. it is suggested to slightly elaborate the conclusions for more effective readings.

Reviewer 3 Report

Comments and Suggestions for Authors

This paper presents the effects of oil dispersion on lubricating film thickness generation under oil droplets supply conditions. The topic is interesting.

This reviewer has some comments.

1.      In figure 1, will the lubricating oil drop due to the influence of gravity since the camera is installed on the testing machine?

2.      In section 3.4, how to distinguish casual and orderly oil supply? Do you have any relevant standards?

3.     In line 300, how to realize the same volume oil supplied at a track with 1 mm width? Please give more details.

4.     The caption of figure 12, how to understand “volume = 0.01 μL, 5P4E”?

5.     In line 326, this reviewer failed to find Table 32, please check carefully  before submitting.

Round 2

Reviewer 3 Report

Comments and Suggestions for Authors

No comments